# Occurrence and Characterisation of Colistin-Resistant *Escherichia coli* in Raw Meat in Southern Italy in 2018–2020

**DOI:** 10.3390/microorganisms10091805

**Published:** 2022-09-08

**Authors:** Gaia Nobili, Gianfranco La Bella, Maria Grazia Basanisi, Annita Maria Damato, Rosa Coppola, Rachele Migliorelli, Valeria Rondinone, Pimlapas Leekitcharoenphon, Valeria Bortolaia, Giovanna La Salandra

**Affiliations:** 1Istituto Zooprofilattico Sperimentale della Puglia e della Basilicata, 20 Manfredonia Street, 71121 Foggia, Italy; 2National Food Institute, Technical University of Denmark, 2800 Kgs. Lyngby, Denmark; 3Department of Bacteria, Parasites and Fungi, Statens Serum Institut, 2300 Copenhagen, Denmark

**Keywords:** *Escherichia coli*, meat, antimicrobial resistance, multidrug resistance, whole genome sequencing, *mcr*, IncX4, MLST

## Abstract

Colistin is a last-resort drug for the treatment of infections by carbapenem-resistant *Enterobacteriaceae*, and the emergence of colistin resistance poses a serious clinical challenge. The aim of this study was to investigate the occurrence of colistin-resistant *Escherichia coli* in retail meat in Southern Italy in 2018–2020. Of 570 samples, 147 contained *E. coli*. Two out of 147 (1.4%) *E. coli* showed a non-wild-type phenotype to colistin and harboured *mcr-1*. *mcr-1* was also detected in a wild-type isolate, resulting in a 2% *mcr* prevalence. *mcr-1*-positive isolates originated from turkey meat collected in Apulia (n = 2) and Basilicata (n = 1). A whole-genome sequencing analysis confirmed *mcr-1.2* and *mcr-1.1* in two and one isolate, respectively. The strains were diverse, belonging to three multi-locus sequence types (ST354, ST410, SLV of ST10) and harbouring genes mediating resistance to antimicrobials in two, six and seven classes. *mcr-1* was carried by IncX4 plasmids with high nucleotide similarity to IncX4 plasmids harbouring *mcr-1.2* and *mcr-1.1* in *Enterobacterales* from different sources and geographical regions. This is the first study reporting updates on *E. coli* non-wild-type to colistin from retail meat in Southern Italy, highlighting the importance of phenotypic and genotypic antimicrobial resistance surveillance to contain the dissemination of *mcr* among *E. coli*.

## 1. Introduction

Antimicrobial resistance (AMR) is a major public health concern with an important impact in the field of animal health and food safety. Antimicrobial-resistant bacteria of livestock origin might reach humans by direct contact and, indirectly, via the consumption of contaminated animal products and water [1,2,3]. Retail meat is considered an important source of bacteria causing food-borne infections and a potential source of antimicrobial-resistant bacteria that may disseminate in the community [4,5]. Colistin is a polymyxin classified among the Highest Priority Critically Important Antimicrobials for human medicine by the World Health Organization (WHO) [6], and it is considered a last resort antimicrobial for the treatment of infections by carbapenem-resistant *Enterobacterales* in humans [7]. Colistin has been widely used in agricultural production and veterinary medicine worldwide for decades, especially for the prevention and treatment of *Enterobacterales* infections [8], which has created a selective pressure for the emergence of colistin resistance.

In 2015, the mobile colistin resistance gene *mcr-1*, a phosphoethanolamine transferase gene located on a transferable plasmid, was discovered in *E. coli* from animals, meat and diseased humans in China [9]. Since then, *mcr-1* has been detected in various bacterial species (e.g., *Escherichia coli*, *Salmonella enterica, Klebsiella pneumoniae, Escherichia fergusonii, Kluyvera ascorbata, Citrobacter braakii, Cronobacter sakazakii, and Klebsiella aerogenes*) from different sources (animals, food products, humans and the environment) in at least in 61 countries [10,11,12,13,14,15]. To date, 10 *mcr* genes and numerous alleles have been reported in *Enterobacterales*, with *mcr-1* being the most frequently detected [10,12,13,14,15,16,17,18,19,20]. The bacteria that carry these genes have been isolated from poultry, pigs, cattle, and food products derived from those animals, but also in human clinical isolates. The spread of *mcr-1* has been associated with various types of plasmids, including IncI2 and IncX4, as the most preventable plasmid types, but also IncF, IncHI1, IncHI2, IncP and IncY [21]. Plasmid-mediated colistin resistance in *Enterobacterales* has also been detected in retail meat, particularly in chicken and turkey meat [22,23].

No recent data are available regarding the prevalence of non-wild-type *E. coli* to colistin in retail meat in Apulia and Basilicata, two regions in Southern Italy that are inhabited by approximately 8% of the resident population in the country (ISTAT data available on https://www.istat.it/en/, accessed on 13 July 2022), and which are visited by over 15 million national and international tourists annually (ISTAT data available on https://www.istat.it/en/, accessed on 13 July 2022). As Italy is a country with an endemic situation of carbapenem-resistant *Enterobacterales* [24], colistin remains an option to safeguard, and possible sources of colistin resistance should be identified to mitigate the risk of transfer to human pathogenic bacteria.

This study aimed at investigating the occurrence and diversity of colistin non-wild-type *E. coli* isolates from retail meat purchased in Apulia and Basilicata in 2018–2020.

## 2. Materials and Methods

### 2.1. Sampling

During 2018–2020, a total of 570 fresh meat samples were randomly collected from supermarkets and butcher stores of Apulia (n = 398) and Basilicata (n = 172) and transferred under refrigerated conditions to the laboratories of Istituto Zooprofilattico Sperimentale della Puglia e della Basilicata (IZS PB, Foggia, Italy). The samples were stored at 4 °C and analysed within 24 h. Out of 570 samples, 142 were chicken meat (102 from Apulia and 40 from Basilicata); 94 were turkey meat (71 from Apulia and 23 from Basilicata); 112 were pork (78 from Apulia and 34 from Basilicata); 133 were beef (86 from Apulia and 47 from Basilicata); and 89 were sheep meat (62 from Apulia and 27 from Basilicata).

### 2.2. Detection and Identification of E. coli

Twenty-five grams of fresh meat sample was homogenized for 2 min with 225 mL of buffered peptone water (Microbiol s.r.l., UTA (CA), Italy). β-glucuronidase-positive *Escherichia coli* were detected by incubating 1 mL of the samples in tryptone bile-glucuronic medium (TBX) at 44 ± 1 °C for 24 h and enumerated according to ISO 16649-2:2001 [25]. Three colonies with typical *E. coli* morphology were randomly selected from each sample. Species identification was performed using MALDI-TOF MS (Matrix Assisted Laser Desorption Ionization—Time of Flight Mass Spectrometry) on one randomly selected *E. coli*-like colony per sample. In case the first isolate was not an *E. coli*, the second and eventually the third isolate were tested. Verified *E. coli* isolates were stored in 60% glycerol at −80 °C.

### 2.3. Multiplex PCR for Detection of Plasmid-Mediated Colistin Resistance Genes

The presence of *mcr-1, mcr-2, mcr-3, mcr-4* and *mcr-5* was determined by a PCR in all *E. coli*, independent of the colistin susceptibility profile. Multiplex PCR was performed according to the protocol of the European Union Reference Laboratory for Antimicrobial Resistance (EURL-AR, Technical University of Denmark, National Food Institute, Denmark) [26].

### 2.4. Antimicrobial Susceptibility Testing

The minimum inhibitory concentration (MIC) of colistin was determined for all verified *E. coli* by the broth microdilution method recommended by EUCAST [27]. Microtitration plates (MERLIN Diagnostika GmbH, Germany) were designed to test 11 concentrations of colistin: 0.0625, 0.125, 0.25, 0.5, 1, 2, 4, 8, 16, 32, 64 mg/L.

Antimicrobial susceptibility profiles were determined for colistin non-wild-type *E. coli* isolates and/or *mcr*-positive *E. coli* using E-Test (bioMérieux Italia Spa, Bagno a Ripoli (FI), Italy), according to the manufacturer’s recommendations. The antimicrobials listed in Panel 1 of the Commission Implementing Decision 2013/652/EU [28] were used for testing, with the exception of colistin.

The obtained MIC values were interpreted in accordance with the interpretative criteria (epidemiological cut-off values—ECOFFs) of the latest updates from EUCAST (European Committee on Antimicrobial Susceptibility Testing) [26], which were supplemented with interpretive criteria from the EFSA Technical Report 2021 [28,29]. *E. coli* strain ATCC25922 was used as a quality control strain. Multidrug resistance (MDR) was defined as the resistance to compounds of at least three antimicrobial classes [30].

### 2.5. Whole-Genome Sequencing

The genomic DNA of colistin resistant and/or *mcr*-positive *E. coli* was extracted using the DNeasy Blood and Tissue Kit (Qiagen, Hilden, Germany), and libraries were prepared with the Nextera XT Kit (Illumina, San Diego, CA, USA), followed by 2 × 250 bp paired-end MiSeq sequencing (Illumina). Raw reads were processed through a quality control (QC) pipeline, as described here: https://bitbucket.org/genomicepidemiology/foodqcpipeline/, accessed on 3 August 2022. In brief, the reads were trimmed using bbduk2 (https://jgi.doe.gov/data-and-tools/bbtools/, accessed on 3 August 2022) to a phred score of 20; reads less than 50 bp were discarded, adapters were trimmed away and a draft de novo assembly was created using SPAdes.21 [31]. From the assemblies, contigs below 500 bp were discarded. The most important parameters that were used to assess the quality of sequencing data were: the number of bases left after trimming, N50, the number of contigs and the total size of assembly. QC parameters used as guidelines were a read depth of at least 25×, N50 of >30,000 bp and a limit on the number of contigs to <500.

Assemblies were analysed with the MLST v.2.0 [32], PlasmidFinder v.2.0 [33] and ResFinder v.4.1 [34] tools available at the Center for Genomic Epidemiology (http://www.genomicepidemiology.org/, accessed on 3 August 2022). The tools were run using default parameters. The contigs harbouring the colistin resistance gene were compared between the isolates using blastn [35], and nucleotide BLAST at NCBI was used to determine the level of identity with publicly available plasmid sequences.

## 3. Results

### 3.1. Microbial Count

The average CFU count of *E. coli* in the analysed samples varied between 2 and 5 × 10^3^ CFU/g, but at levels below 10^2^ CFU/g in most of them (approximately 92%).

From 570 beef, pork, ovine, turkey and chicken retail meat samples, 25.8% contained *E. coli* (Table 1), which was confirmed by using MALDI-TOF technology.

### 3.2. Multiplex PCR for mcr-1, mcr-2, mcr-3, mcr-4 and mcr-5

The *mcr-1* was detected in three (2%) out of 147 isolates tested. In two of these isolates, the presence of *mcr-1* was concordant with the detected non-wild-type phenotype to colistin, whereas the third *mcr-1*-positive isolate (ID 506) exhibited a colistin wild-type phenotype (MIC = 0.25 mg/L). *mcr-1*-positive isolates originated from Apulia (n = 2) and Basilicata (n = 1) in 2018 (n = 2) and 2020 (n = 1). *mcr-2*, *mcr-3*, *mcr-4* and *mcr-5* were not detected in any isolate.

### 3.3. Antimicrobial Susceptibility Testing

Two out of 147 (1.4%) *E. coli* (ID 53 and ID 67) showed a non-wild-type phenotype to colistin (MIC > 2 mg/L). These strains originated from turkey meat samples. All other isolates were colistin wild-type.

Antimicrobial susceptibility testing was performed on *mcr-1*-positive *E. coli* isolates, and the results are reported in Table 2. The three isolates showed resistance to compounds from two, six and seven antimicrobial classes (polymixins, β-lactams, tetracyclines, quinolones, sulphonamides, phenicols, macrolides). Isolates 53 and 506 showed multidrug (MDR) resistance profiles.

### 3.4. Whole-Genome Sequencing

The Center for Genomic Epidemiology (CGE) ResFinder Tool confirmed the presence of the *mcr-1.2* in two isolates (ID 53 and 67) and *mcr-1.1* in one isolate (ID 506).

These isolates belonged to three multi-locus sequence types (MLST) (ST354, ST410, SLV of ST10 due to a new adk allele) and harboured different plasmid replicons and antimicrobial resistance genes (Table 2). The AMR genotypes were in agreement with the observed resistance phenotypes, except for the *mcr*-positive, colistin susceptible isolate mentioned above. The *mcr* genes were harboured by IncX4 plasmids. The contig with IncX4 and *mcr-1.2* measured 32,737 bp and 30,058 bp in *E. coli* ID 53 and 67, respectively, whereas the contig with IncX4 and *mcr-1.1* measured 29,192 bp in *E. coli* ID 506.

The two contigs with IncX4 and *mcr-1.2* were nearly identical, as they only had two single nucleotide variants over 30,058 bp. The contig with IncX4/mcr-1.1 (*E. coli* ID 506) compared with both IncX4/*mcr-1.2* contigs showed a 99.9% nucleotide identity over 29,000 bp with *E. coli* ID 53 and over 26,553 bp with *E. coli* ID 67.

By using the *mcr-1.2*-positive contig from *E. coli* ID 53 for blastn, 94 hits with high identity (99–100%) over the entire sequence were obtained. Similar results were obtained by using *mcr-1.2*-positive contig from *E. coli* ID67 (91 hits with 99–100% identity over 100% of the sequence) and mcr-1.1-positive contig from *E. coli* ID 506 (100 hits with 99–100% identity over 100% of the sequence).

These hits referred to IncX4 plasmids harbouring *mcr-1.2* or *mcr-1.1* detected in different *Enterobacteriaceae* (*E. coli*, *Salmonella* sp. and *Klebsiella pneumoniae*) from different sources (humans, food, wastewater) and different geographical regions including, among others, Italy and Switzerland.

The sequence data obtained in this study have been deposited in the European Nucleotide Archive (ENA) at EMBL-EBI under accession number PRJEB47231 (*E. coli* ID 53 and *E. coli* ID 67) and PRJEB49095 (*E. coli* ID 506).

## 4. Discussion

In this study, we first investigated the occurrence of colistin-resistant *E. coli* in a large number of samples (n = 570) of different types of retail meat purchased in the Southern Italy regions of Apulia and Basilicata between 2018 and 2020. The results revealed a prevalence of 3.2% (3/94) of turkey meat samples contaminated with colistin-resistant *E. coli*, whereas 89, 112, 133 and 143 samples of ovine, pork, beef and chicken meat were negative for colistin-resistant *E. coli*. The recovery of *E. coli* in the various meat types was in the expected range in terms of both the prevalence of *E. coli*-contaminated samples, ranging from 8.2% of positive pork sample to 27.9% of positive turkey meat samples, and in terms of CFU counts that varied between 2 and 5 × 10^3^ CFU/g, which supports that the observed rare to low occurrence of colistin-resistant *E. coli* reflects the true situation and good hygiene practices of different food production steps. Another recent study performed in six broiler and two turkey commercial flocks in Italy showed that *mcr* genes were not detected in 38 litter samples [36]. These results are in agreement with the data obtained through the harmonised monitoring of AMR in zoonotic and commensal bacteria from food-producing animals and meat thereof performed yearly by the EU Member States in accordance with EU legislation (Directive 2003/99/EC17 and Commission implementing Decision (EU) 2020/1729) [29,37,38]. According to the EFSA Report 2021, summarising monitoring data from 2019 to 2020, median levels of colistin resistance for all reporting countries ranged from rare in isolates from pigs and broilers to very low and low in isolates from calves and turkeys, respectively [39]. The same report also showed that, in Italy, colistin resistance decreased significantly in indicator *E. coli* from broilers and turkeys, and remained at low levels in indicator *E. coli* from cattle and pigs [40], in the period 2014–2020.The low occurrence of colistin-resistant *E. coli* in animals and meat thereof in EU countries is likely a consequence of the reduction in the use of colistin in farms due to European policies [40].

The turkey meat samples contaminated by *mcr-1*-positive *E. coli* were from different turkey flocks and farms and were collected from two supermarkets in different provinces of Apulia region (Foggia and Lecce) and one supermarket in a province of Basilicata region (Matera) in 2018 (n = 2) and 2020 (n = 1). The three *mcr-1*-positive *E. coli* were phenotypically and genetically diverse and harboured *mcr-1.1* or *mcr-1.2* on IncX4 plasmids that were highly similar over a 26,553 bp region, at least. In one strain, only the genetic analysis of colistin resistance allowed the identification of *mcr-1.1*, as the isolate exhibited a colistin-susceptible phenotype (MIC = 0.25 mg/L). The detection of *mcr* genes in colistin wild-type isolates has been reported previously [41,42,43,44,45,46] as a consequence of the absence of insertion sequences in association with *mcr-1.1* [42,46]. This study raises a concern that ‘silent’ *mcr-1* genes have the potential to disseminate and eventually compromise the treatment of infections by carbapenem-resistant *Enterobacterales*. The accumulation of antimicrobial resistance determinants in two (67%) of the *mcr*-positive strains is notable. *E. coli* ID53 and *E. coli* ID506 were MDR, as they exhibited resistance to compounds in up to seven antimicrobial classes (Table 2). The wide diffusion of resistance to tetracycline, sulphonamides and ampicillin is probably attributable to antimicrobial usage in farms [47], as these antimicrobials represent the most commonly used drugs in food-producing animals [48]. Thus, *mcr-1* positive strains should be considered not only as reservoirs of colistin resistance genes, but also of genes conferring resistance to additional antimicrobials. 

The genomic characteristics of the isolates suggest that *mcr-1* spread from a specific type of plasmid (IncX4) through genetically different *E. coli* strains belonging to three MLST types. The IncX4 plasmid is considered one of the main carriers of the *mcr-1* gene in *Enterobacteriaceae* [43,44,45,46,49]. IncX4 plasmids are usually 33,000–35,000 bp in size and conjugative [44] and are one of the major groups found in *E. coli* isolates, but also described in other *Enterobacterales* species [50,51]. BLAST at the NCBI of the 29,192–32,737 bp IncX4/*mcr-1*-positive contigs clearly showed that the three strains in this study harboured an epidemic IncX4/*mcr-1.2* or IncX4/*mcr-1.1* plasmid that was able to spread across the isolates of different species.

## 5. Conclusions

As only a few data regarding *mcr*-positive *E. coli* from animal sources were available in Italy prior to this study, and Italy has been a country in which colistin has been widely used in farms, this study was an attempt to improve the knowledge on the occurrence and diversity of colistin-resistant *E. coli* in retail meat in Italy. The present study confirms that poultry meat is generally at a higher risk of contamination by *mcr-1*-positive *E. coli* compared to other meat types, in agreement with previous data from Italy [23] and other countries such as Germany, France, Poland and Czechia [22,40,52,53]. In particular, retail turkey meat seems to be the main source of *mcr-1*-carrying *E. coli* compared to other meat types, possibly as a consequence of both the longer life span of turkeys compared to chickens—which implies a longer length of exposure to selective pressure favouring antimicrobial resistance [3]—and of lower hygienic standards and a higher risk of carcass contamination in poultry slaughtering compared to cattle and pig slaughtering. These results highlight the importance of AMR monitoring plans in turkey meat in Italy, which is the fifth poultry meat producer in the EU (10% of EU total production in 2021) [54]. The fact that the *mcr-1* gene was localised on an IncX4 plasmid that could spread across different colistin-resistant and colistin-susceptible *E. coli* detected in different geographical areas and in different years shows that attention towards *mcr*-positive *E. coli* in animal sources should not diminish yet, as epidemic *mcr-1* plasmids are still circulating in *E. coli* populations in animals and meat. However, at the same time, this study also suggests that measures to reduce the use of colistin in production animals in Italy have been effective in mitigating the risk of meat contamination by colistin-resistant *E. coli*.

## Figures and Tables

**Table 1 microorganisms-10-01805-t001:** Numbers of *E. coli* and *E. coli mcr*-positive isolated from retail meats in Apulia and Basilicata.

Category	No. of Samples	No. of Isolates (%)	
		*E. coli*	*E. coli mcr positive*
**Beef**	133	17 (11.6)	0 (0.0)
**Pork**	112	12 (8.2)	0 (0.0)
**Ovine**	89	21 (14.3)	0 (0.0)
**Chicken**	142	56 (38)	0 (0.0)
**Turkey**	94	41 (27.9)	3 (3.2)
**Total**	570	147 (25.8)	3 (0.5)

**Table 2 microorganisms-10-01805-t002:** Genetic and phenotypic traits of *mcr-1*-positive *Escherichia coli*.

Isolate	MLST	Plasmid Finder	AMR Genes	Antimicrobial Class	AMR Phenotype
**53**	354	Col (MG828), Col8282, IncFIB (AP001918), IncFIC(FII), IncI1-I(Gamma), IncX4	*mcr-1*.2	Polymixins	Colistin
			*gyrA* S83L, *gyrA* D87N, *parC* S80I, *parC* E84G, *parE* I355T	Quinolones	Nalidixic acid Ciprofloxacin
			*aph(6)-Id* *aph(3″)-Ib, aadA1*	Aminoglycosides	(Streptomycin, predicted)
			*dfrA1* *sul1* and *sul2*	Sulphonamides	Trimethoprim Sulfamethoxazole
			*tet(A)*	Tetracyclines	Tetracycline
			*bla_TEM-1b_*	β-Lactams	Ampicillin
**67**	^1^ SLV of ST10	Col440I, IncFIB (AP001918), IncFII(pSE11), IncI2, IncX1, IncX4	*mcr-1*.*2*	Polymixins	Colistin
			*gyrA* S83L *gyrA* D87N *parC* S80I	Quinolones	Nalidixic acid Ciprofloxacin
**506**	ST410 CplxST23	Col (MG828), Col440I, ColRNAI, IncFIA, IncX4	*mcr-1.1*	Polymixins	Colistin susceptible
			*gyrA* D87N *parC S*80I	Quinolones	Nalidixic acid Ciprofloxacin
			*dfrA1* *sul2*	Sulphonamides	Trimethoprim Sulfamethoxazole
			*catA1*	Phenicols	Chloramphenicol
			*tet(B)*	Tetracyclines	Tetracycline
			*bla_TEM-1b_*	β-Lactams	Ampicillin
			*mph(A)*	Macrolides	Azithromycin

^1^ SLV, single locus variant. The adk allele is new.

## Data Availability

The data presented in this study are available on request from the corresponding author.

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
