# Peer review of "Occurrence and Characterisation of Colistin-Resistant Escherichia coli in Raw Meat in Southern Italy in 2018–2020"

_microorganisms, 2022, doi:10.3390/microorganisms10091805_

Round 1

Reviewer 1 Report

Its good paper but needs some changes from authors

1-   English language needs improve by professionals 

2-   Why authors isolated E.coli and ignore another species?

3-   Why they did not used genetic methods for detection E.coli?

4-   Please put this part (Colistin Susceptibility Testing) and (Antimicrobial Susceptibility Testing) in the same paragraph.

5-    In methods section Multiplex PCR for detection of plasmid-mediated colistin resistance genes, so where the table of primers?

6-   Why the authors used Whole-genome sequencing and detection colistin resistance by PCR? I think WGS good tools for analysis genes of colistin !!

7-   Why did not detection another type of antibiotics and determine if this isolates MDR or just colistin resistance?

8-   In 2.5. Antimicrobial Susceptibility Testing paragraph, the authors said (MDR) was defined as resistance to compounds of at least three antimicrobial classes did they detection that?

9-   How many type of antibiotics used for detection Antimicrobial Susceptibility Testing by E-test? Please listed in methods in Antimicrobial Susceptibility Testing.

10-               Did you used study in Middle East? As compare with your results? Also please use research from Middle East or review about colistin in reference if you can.

Author Response

Response to Reviewer 1 Comments

Point 1: English language needs improve by professionals 

Response 1: The Authors wish to thank the reviewer for the time spent reading our paper and for the useful suggestions that helped improving the manuscript. Thank you for the appreciation of our work.

The manuscript has been reviewed by a native speaker.

Point 2: Why authors isolated E.coli and ignore another species?

Response 2: The mcr genes have been reported worldwide in Enterobacterales (including E. coli, Salmonella, and K. pneumoniae) from various sources, especially the environment and animals [Elbediwi et al., 2019]. E. coli is the most prevalent species among the mcr-harboring isolates reported so far, accounting for approximately 91% of the total mcr-carrying isolates [Nang et al., 2019].

Therefore, the authors decided to evaluate the occurrence of colistin-resistant Escherichia coli in raw meat in this study. Other species of microorganisms will be considered in further studies.

Point 3: Why they did not used genetic methods for detection E.coli?

Response 3: The authors were interested in microbiological isolation in order to carry out a thorough characterization of the microorganisms and to study antimicrobial susceptibility profile.

Point 4: Please put this part (Colistin Susceptibility Testing) and (Antimicrobial Susceptibility Testing) in the same paragraph.

Response 4: Thank you for your suggestions. The revisions have been inserted into the manuscript

Point 5: In methods section Multiplex PCR for detection of plasmid-mediated colistin resistance genes, so where the table of primers?

Response 5: Multiplex PCR was performed according to the protocol of the European Union Reference Laboratory for Antimicrobial Resistance (EURL-AR, Technical University of Denmark, National Food Institute, Denmark) (Rebelo et al., 2018), where primers used are listed (Table 1).

Point 6: Why the authors used Whole-genome sequencing and detection colistin resistance by PCR? I think WGS good tools for analysis genes of colistin!!

Response 6: The authors agree with you about the importance of WGS for the molecular characterization of antimicrobial resistance genes.

In this study, the presence of mcr genes was determined by PCR in all E. coli, independent of the colistin susceptibility profile, while WGS analysis was performed on colistin-resistant strains and mcr-positive E. coli.

Point 7: Why did not detection another type of antibiotics and determine if this isolates MDR or just colistin resistance?

Response 7: In this study, colistin susceptibility was evaluated on all E. coli by the broth microdilution method. Subsequently, colistin non-wild type E. coli and/or mcr-positive E. coli were tested for susceptibility to the antimicrobials listed in Panel 1 of the Commission Implementing Decision 2013/652/EU using E-Test. Results are presented in the section 3.3 and table 2.

Point 8: In 2.5. Antimicrobial Susceptibility Testing paragraph, the authors said (MDR) was defined as resistance to compounds of at least three antimicrobial classes did they detection that?

Response 8: The authors reported that E. coli ID53 and E. coli ID506 were MDR as they exhibited resistance to compounds in up to seven antimicrobial classes (see Table 2).

Point 9: How many type of antibiotics used for detection Antimicrobial Susceptibility Testing by E-test? Please listed in methods in Antimicrobial Susceptibility Testing.

Response 9: In the manuscript, the authors reported the reference of the Commission Implementing Decision 2013/652/EU in which the antimicrobials were listed in Panel 1. All antimicrobials were tested using E-Test, with the exception of colistin.

Point 10: Did you used study in Middle East? As compare with your results? Also please use research from Middle East or review about colistin in reference if you can.

Response 10: As only few data regarding mcr-positive E. coli from animal sources were available in Italy prior to this study, and Italy has been a country in which colistin has been widely used in farms, this study was an attempt to improve the knowledge on occurrence and diversity of colistin-resistant E. coli in retail meat in Italy. In addition, our results were compared to those inserted in EFSA Report 2021.

Reviewer 2 Report

Few data regarding mcr-positive E. coli from animal sources have been reported in Italy so far. I found this topic of scientific interest; this kind of surveillance is important to develop strategies to address the drug resistance problem and to avoid dissemination of mcr among E. coliThe manuscript is well written and the results will contribute to the current knowledge in the field!

Line 209 - therefore

Author Response

Authors wish to thank you for the time spent at reading our paper and for the useful suggestions that helped in improving the manuscript. Thank you for the appreciation of our work.